# SWE-Effi: Re-Evaluating Software AI Agent System Effectiveness Under Resource Constraints

## Abstract

The advancement of large language models (LLMs) and code agents has demonstrated significant potential to assist software engineering (SWE) tasks, such as autonomous issue resolution and feature addition. Existing AI for software engineering leaderboards (e.g., SWE-bench) focus solely on solution accuracy, ignoring the crucial factor of effectiveness in a resource-constrained world. This is a universal problem that also exists beyond software engineering tasks: any AI system should be more than correct—it must also be cost-effective. To address this gap, we introduce SWE-Effi, a set of new metrics to re-evaluate AI systems in terms of holistic effectiveness scores. We define effectiveness as the balance between the accuracy of outcome (e.g., issue resolve rate) and the resources consumed (e.g., token and time). In this paper, we specifically focus on the software engineering scenario by re-ranking popular AI systems for issue resolution on a subset of the SWE-bench benchmark using our new multi-dimensional metrics. We found that AI system's effectiveness depends not just on the scaffold itself, but on how well it integrates with the base model, which is key to achieving strong performance in a resource-efficient manner. We also identified systematic challenges such as the "token snowball" effect and, more significantly, a pattern of "expensive failures". In these cases, agents consume excessive resources while stuck on unsolvable tasks—an issue that not only limits practical deployment but also drives up the cost of failed rollouts during RL training. Lastly, we observed a clear trade-off between effectiveness under the token budget and effectiveness under the time budget, which plays a crucial role in managing project budgets and enabling scalable reinforcement learning, where fast responses are essential. [1]

## 1 Introduction

Leaderboards for benchmarks like SWE-bench Jimenez et al. (2024) provide a foundation for measuring the progress of AI coding assistants in repository-level SWE (software engineering) tasks, such as resolving issues submitted by developers. Yet, the spotlight on these leaderboards shines almost exclusively on a single metric: the resolve rate. Current evaluation paradigms often operate on the implicit assumption of unlimited resources, but companies and developers live in a resource-constrained reality. Was the solution found in minutes, or did it burn through hours of expensive GPU time? Can a team even afford to run this agent on a daily basis? By ignoring such questions, current benchmarks create a disconnect from the realities of practical AI deployment. While our focus is on software engineering, this is a universal problem: a solution from any AI system should be more than correct—it must also be cost-effective.

This perspective shift is critical for two key frontiers in AI-driven software development. First, a trend emerges where agents consume massive LLM tokens (e.g., by test-time compute) to resolve one particular SWE task and earn marginal gains on leaderboards. This raises a critical question about the law of diminishing returns: Is a 1% improvement in resolve rate worth a 5x increase in cost? We argue that this path may not be the most promising direction for building truly scalable and accessible tools. Second, SWE agent scaffolds are increasingly being used as foundations by research

---

[1]We open source all experiment data, code, and public leaderboard.

teams to train smarter, self-improving models that specialize in SWE tasks through Reinforcement Learning (RL). In long-horizon RLs (like DeepResearch Agent Zhang et al. (2025) and SkyRL Cao et al. (2025) training), they usually have effectiveness requirements so that the RL process will not run forever. In the SWE task scenario, each RL training step, or "rollout," involves setting up the project environment, LLM reasoning, code execution, and test validation. If a SWE agent scaffold is slow or expensive, it hinders the entire RL process. High latency per rollout becomes a significant barrier to effective RL training, limiting progress across the community. For instance, DeepSWE Luo et al. (2025) developed a custom scaffold to ensure low-latency performance for RL. The Sky-RL Cao et al. (2025) project uses OpenHands Wang et al. (2025) but requires extensive optimizations to make training feasible. Similarly, SWE-RL Wei et al. (2025) employs the lightweight Agentless-Mini Wei et al. (2025) yet avoids direct execution feedback in RL, likely due to high execution costs. These challenges arise because RL relies on massive-scale trial and error, directly affected by the scaffold's performance.

To bridge this gap, we introduce the SWE-Effi, a set of new metrics to re-evaluate AI systems in terms of holistic effectiveness scores. We systematically re-evaluate and analyze AI systems[2] on top of the well-established SWE-bench. To be clear, SWE-Effi is designed to complement existing benchmarks like SWE-bench. SWE-Effi offers a holistic perspective on AI system evaluation and highlights performance under resource constraints, emphasizing effectiveness under the cost budget. Moreover, SWE-Effi serves as a valuable indicator for identifying promising foundations in Reinforcement Learning for SWE tasks.

Our re-evaluation of popular AI systems through this lens yields critical, and often counter-intuitive, insights. We find that a system's effectiveness is not an inherent property of its scaffold but emerges from its synergy with the base LLM. For instance, while the SWE-Agent scaffold achieves a respectable Effectiveness under Token Budget (EuTB) score of 21.8% with the Qwen3-32B model, its effectiveness collapses when paired with GPT-4o-mini, causing the EuTB score to a mere 5.1%. This dramatic drop is symptomatic of a larger issue we identify: "expensive failures." Our analysis reveals that for this same system, an unresolved attempt consumes on average over 4 times more resources than a successful one. These hidden costs, completely invisible to traditional resolve-rate metrics, demonstrate the critical need for our holistic evaluation paradigm.

Our primary contribution is a new evaluation perspective, metrics, and a public leaderboard. We conducted an exploratory study, which analyzed five well-known SWE agent scaffoldings (AutoCodeRover Zhang et al. (2024), SWE-agent Yang et al. (2024), OpenHands Wang et al. (2025), Agentless Xia et al. (2024), Agentless-Mini Wei et al. (2025)) paired with three different LLMs (GPT-4o-mini, Llama-3.3-70B, Qwen3-32B) on a representative subset of SWE issues. While we tracked the final resolve rate, our main focus was on evaluating the overall AI system's time and resource effectiveness in resolving issues and analyzing the characteristics behind it. The findings we present are not intended to be a final, definitive verdict on these five scaffolds. Rather, they serve as an example of the deeper understanding we can unlock by looking at the same problems from a different perspective with our lightweight evaluation.

## 2 RELATED WORK

### 2.1 AI SYSTEM EVALUATION ON SOFTWARE ENGINEERING TASKS

With the development of LLMs, researchers have begun to investigate whether LLMs can perform more complicated software engineering tasks beyond direct code generation. Jimenez et al. (2024) first introduced SWE-bench to evaluate LLMs' ability to resolve real GitHub issues, which typically involves understanding the issue, localizing files to be edited, and modifying them. Follow up work SWE-bench-Verified OpenAI (2024) then helped to improve quality and robustness of the benchmark through manual evaluation and selection of high-quality instances by human experts. Badertdinov et al. (2025) also extended SWE-bench work by providing 21,000 new real-world tasks in order to expand the limited number of high quality SWE data and ensure better evaluation on SWE tasks. At the same time, Zan et al. (2025) and Guo et al. (2025) proposed SWE task evaluations with more programming languages as well as including multimodal data such as diagrams and images. Beyond

---

[2]By "AI system" we refer to a single software system that includes an AI model (LLM) with a software scaffold (e.g., agent) working together to solve a given task.

GitHub issue resolution, there have been numerous works aimed at using LLMs for various aspects of software engineering. Mündler et al. (2024) proposed SWT-BENCH to evaluate LLM coding agents' ability to generate test cases that reproduce issues described in natural language, meanwhile, He et al. (2025) proposed SWE-Perf, a benchmark to evaluate LLM proficiency in improving code performance and optimization. Zhao et al. (2024) introduced GTArena to evaluate multimodal LLM agents for end-to-end automated GUI testing. Zhu et al. (2025) proposed CVE-Bench to challenge LLM agents to exploit vulnerable web applications. However, despite these extensive benchmarks evaluating LLM and agent performance, none of them systematically measure the efficiency of the SWE AI systems, often only focusing on the correctness of the generated solutions.

## 2.2 LLM-based methods for GitHub Issue Resolution

In recent years, resolving GitHub issues with LLMs has been an extraordinarily active research topic Yang et al. (2024); Zhang et al. (2024); Antoniades et al. (2024); Xia et al. (2024); Yang et al. (2025b); Tao et al. (2024); Su et al. (2025); Wang et al. (2025); Gao et al. (2025). Following recent surveys Dong et al. (2025); Yang et al. (2025a), these methods can be categorized into two types: *agentic* methods and *procedural* methods. Agentic methods resolve issues by using LLM to independently plan, make decisions, and use tools to complete a task. In this paradigm, Yang et al. (2024) proposed SWE-Agent that equips LLM with basic editing and testing tools in a ReAct Yao et al. (2023) loop, allowing the agent to reason, observe, and interact with the environment by employing a set of tools. OpenHands Wang et al. (2025) augmented LLMs with a more extensive toolset, it implements an interaction mechanism that allows human to be involved in the ReAct loop. Procedural methods treat issue resolution as a fixed pipeline that invokes LLMs at scripted checkpoints in the process. For instance, AutoCodeRover Zhang et al. (2024) localizes suspicious functions by searching the codebase with a set of AST-based tools, then generates patches based on LLM's understanding. Agentless Xia et al. (2024) uses hierarchical localization to identify edited code, then creates several candidate patches and chooses one based on test results. While there is no consensus on which approach is better, the performance and resource efficiency of a scaffold are highly coupled with the strengths and weaknesses of the underlying LLM. To guide the selection of approaches under various budget constraints, a systematic evaluation of their multifaceted efficiency is necessary, and is missing from previous research.

## 3 The SWE-Effi Resource Effectiveness Evaluation Metrics

To address the limitations of traditional, single-metric evaluations, we introduce **SWE-Effi**, a multi-dimensional framework designed to provide a holistic assessment of AI systems for software engineering. The framework is structured in two layers: a foundational layer of **Core Performance Metrics**—the direct, raw measurements from each experimental trial—and a higher-level layer of **Resource Effectiveness Metrics**, which are derived scores that evaluate overall efficiency.

### 3.1 Core Performance Metrics: The Foundational Measurements

This first set of metrics represents the fundamental, raw data collected for each trial. These direct measurements serve as the building blocks for our subsequent efficiency analysis.

**Resolve Rate (%)** The primary outcome metric, representing the percentage of issues an AI system successfully resolves.

**CPU Time (s)** The raw computational time consumed by the AI system's scaffold for local operations (e.g., file manipulation, test execution), exclusive of LLM inference latency.

**LLM Calls, Input & Output Tokens** The raw counts of API requests made to the LLM, and the number of tokens sent to (input) and generated by (output) the model during a trial.

**Normalized Inference Time (s)** A standardized, hardware-agnostic measure of LLM-related latency. To ensure fair comparisons, raw wall-clock time is replaced with a value predicted by a linear regression model, as detailed in Section 3.1.1.

### 3.1.1 Derivation of Normalized Inference Time

A key challenge in evaluating systems that rely on external APIs is that raw wall-clock time is an unreliable metric, as it is heavily influenced by external factors such as network conditions and provider-specific hardware. To establish a standardized measure for inference time, we developed a predictive linear regression model. We chose OpenAI's GPT-4o-mini OpenAI as our performance baseline due to its stable and widely representative performance. We trained this model on a dataset of 515,041 individual raw API call logs collected in our experiment in Section 4, partitioned into a 90% training set and a 10% validation set. The model achieved a strong coefficient of determination ($R^2$) of 0.79 on the validation set, indicating a good fit. The resulting equation is defined as:

$$\text{Normalized Inference Time (s)} = \underbrace{1.457}_{\text{Fixed Overhead }(\alpha)} + \underbrace{4.266 \times 10^{-5}}_{\text{Per-Input-Token Latency }(\beta_1)} \times (\text{Input Tokens})$$
$$+ \underbrace{4.999 \times 10^{-3}}_{\text{Per-Output-Token Latency }(\beta_2)} \times (\text{Output Tokens}) \tag{1}$$

The components of this model represent distinct phases of an API call:

- **Fixed Overhead ($\alpha$):** A constant base latency of 1.457 second, accounting for fixed costs like network round-trips and initial request processing.
- **Per-Input-Token Latency ($\beta_1$):** The time cost to process each input token, corresponding to the model's "prefill" phase.
- **Per-Output-Token Latency ($\beta_2$):** The time cost to generate each output token, corresponding to the "decoding" phase.

This validated regression model serves as our universal yardstick for standardizing time. For every LLM call made by any system, we apply Equation 1 to its input and output token counts to calculate its Normalized Inference Time. It translates raw latencies from different services into a single, hardware-agnostic value, enabling a fair and direct comparison across all experiments.

### 3.2 Resource Effectiveness Metrics: Scoring Efficiency

While the core metrics provide raw data points, they do not, by themselves, measure efficiency. For example, a high CPU time is only meaningful when compared to the number of problems solved. To capture this relationship, we introduce our effectiveness metrics. These are derived scores that contextualize resource consumption against task success using the **Area Under the Curve (AUC)**.

For each effectiveness metric, we plot the cumulative number of resolved tasks (derived from the *Resolve Rate*) against the consumption of a specific core metric (e.g., *CPU Time*, *Total Tokens*; example curves are provided in Appendix). The resulting AUC is then normalized to a score between 0 and 1, providing a single, comparable measure of efficiency under a defined budget. Our leaderboard is built on the following four effectiveness scores:

- **Effectiveness under Token Budget (EuTB):** Scores the efficiency of *token usage* by calculating the AUC of the resolve rate vs. total tokens per issue curve, capped at 2 million tokens.
- **Effectiveness under Cost Budget (EuCB):** Scores the efficiency of *monetary expenditure* by calculating the AUC of the resolve rate vs. dollar cost per issue curve, capped at $1.00 USD[3].
- **Effectiveness under CPU Time Budget (EuCTB):** Scores the efficiency of the *local computation time* by calculating the AUC of the resolve rate vs. CPU time per issue, capped at 30 minutes.
- **Effectiveness under Inference Time Budget (EuITB):** Scores the efficiency of the system's usage of *LLM inference time* by calculating the AUC of the resolve rate vs. normalized inference time per issue, also capped at 30 minutes.

By combining this setup with our detailed measurement, we have created a leaderboard that reflects not just success, but the true cost of that success. The following sections present our detailed experiment setup and our initial observations based on the collected data.

---

[3]Per-token costs used are provided in the Appendix 7.1

## 4 EXPERIMENT SETUP

To ensure our leaderboard provides meaningful and fair comparisons, we meticulously selected the baseline SWE scaffolds, LLMs, and controlled the experiment environment as described below.

*SWE Scaffold Selection and Configuration:* We selected five popular, actively maintained open-source SWE scaffolds from the top of the SWE-bench Jimenez et al. (2024) leaderboard (as of May 2025): AutoCodeRover Zhang et al. (2024), OpenHands Wang et al. (2025), SWE-Agent Yang et al. (2024), Agentless Xia et al. (2024), and Agentless-Mini Wei et al. (2025). AutoCodeRover, Agentless, and Agentless-Mini are procedure-based scaffolds that use LLMs to solve software engineering problems in a structured workflow, whereas OpenHands and SWE-Agent are representative agentic-style scaffolds designed for software engineering tasks. They enable LLMs to autonomously plan, edit code, and execute commands by interacting with external tools (e.g., bash), thereby facilitating end-to-end task completion. We configured those scaffolds based on their official guidelines and adhered to their default iteration or generation limits to provide a baseline for "out-of-the-box" performance, except for the termination criteria for SWE-Agent where we adjusted its API budget to a strict $1 per issue. This forces the agent to operate under the same kind of financial pressure a real team would face. To ensure our time duration measurements were accurate and unbiased, we explicitly disabled any parallel processing capabilities for the five scaffolds. This allowed us to measure the true, sequential processing time of each scaffold's core logic, making our time-based comparisons direct and fair. To capture the fine-grained data for our analysis, we augmented each SWE scaffold's logging and added our own profiling code to collect the precise time cost regarding local CPU computation and the backend LLM inference for each AI system.

*LLM Selection*: We selected a subset of LLMs that represent balance of performance and cost, making them suitable for production environments where budgets are a key consideration. To analyze performance across different resource conditions, our selection covers a range of types and sizes:

- GPT-4o-mini-2024-07-18 OpenAI: A proprietary model from OpenAI, designed for high efficiency and speed. It serves as a baseline for performance in cost-sensitive scenarios.
- Llama-3.3-70B-Instruct Meta: A larger, open-source model from Meta. Its 70B parameter size provides strong, general-purpose capabilities and represents a high-performance option within the open-source ecosystem. Note that this model was quantized to FP8 format.
- Qwen3-32B Qwen: A mid-sized, 32B parameter open-source model from Alibaba, noted for its reasoning abilities. We selected it specifically for its strong reasoning abilities relative to its size. This allows us to evaluate the performance of a more lightweight but capable alternative, which is an attractive profile for resource-constrained environments.

To use these models, we accessed GPT-4o-mini via the OpenAI API, Llama-3.3-70B via Together.AI's inference service, and Qwen3-32B via Alibaba Cloud's API. We modified the selected five SWE scaffoldings where necessary to handle specific API behaviors, such as the streaming-only output of Qwen3-32B from Together.AI, to ensure consistent operation.

*Benchmark*: We evaluated all scaffold-model combinations on a focused subset of 50 issues randomly drawn from the well-respected SWE-bench-Verified dataset. This lightweight sample allows us to draw conclusions with 95% confidence within a confidence interval of 13%, aligning with software practitioners' needs in evaluating and selecting a suitable framework with affordable resources and time consumption. To ensure this subset was a fair representation of the whole, we used stratified sampling, preserving the original distribution of issues across different software projects. We will make our subset publicly available on HuggingFace.

## 5 EVALUATION OF AI SYSTEM SWE CAPABILITY AND EFFICIENCY

In this section, we present the main experimental observations for the five selected scaffolds paired with three distinct LLMs described in Section 4. Tables 1 and 2 present the Resource Efficiency (Section 3.2) and Core Performance (Section 3.1) metrics, respectively, which form the basis of our SWE-Effi evaluation. All reported values are aggregated and averaged across all issue instances, including both resolved and unresolved attempts, on our curated subset of the SWE-bench-Verified OpenAI (2024) benchmark. Additionally, we provide our analysis and uncover several crucial trends and

Table 1: Performance comparison of SWE AI systems (Scaffold + LLM). The table reports the final Resolve Rate (%) and our SWE-Effi effectiveness scores (detailed in Section 3). Higher values indicate better performance. We mark the best-performed SWE AI system on a metric as bolded.

| SWE Scaffold | Base Model | EuTB (%) | EuITB (%) | EuCTB (%) | EuCB (%) | Resolve Rate (%) |
|---|---|---|---|---|---|---|
| AUTOCODEROVER | GPT-4o-mini | 11.9 | 11.6 | 12.0 | 10.2 | 12.0 |
| | Llama-3.3-70B | 17.3 | 26.2 | 27.9 | 27.4 | 28.0 |
| | Qwen3-32B | 37.1 | **33.1** | **37.9** | **37.0** | 38.0 |
| OPENHANDS | GPT-4o-mini | 6.8 | 9.5 | 9.7 | 6.1 | 11.9 |
| | Llama-3.3-70B | 17.0 | 19.3 | 19.4 | 19.3 | 20.0 |
| | Qwen3-32B | 22.7 | 30.7 | 32.7 | 26.3 | 34.0 |
| SWE-AGENT | GPT-4o-mini | 5.1 | 8.3 | 9.7 | 2.5 | 10.0 |
| | Llama-3.3-70B | 11.8 | 11.8 | 11.9 | 12.0 | 12.0 |
| | Qwen3-32B | 21.8 | 19.7 | 27.7 | 21.7 | 28.0 |
| AGENTLESS | GPT-4o-mini | 25.3 | 20.5 | 11.9 | 25.5 | 26.0 |
| | Llama-3.3-70B | 26.8 | 21.3 | 22.1 | 27.8 | 28.0 |
| | Qwen3-32B | **46.7** | 29.9 | 30.3 | 47.1 | **48.0** |
| AGENTLESS-MINI | GPT-4o-mini | 19.6 | 19.8 | 20.0 | 19.4 | 20.0 |
| | Llama-3.3-70B | 3.9 | 4.0 | 4.0 | 4.0 | 4.0 |
| | Qwen3-32B | 27.5 | 27.2 | 28.0 | 27.3 | 28.0 |

Table 2: Resource performance metrics comparison of SWE AI systems. Taken as the average result across all issue instances (both resolved and unresolved instances) for SWE-bench-Verified. *Inf.* - normalized inference time, $k$ - thousand. $T_{IN/OUT}$ - input/output tokens, respectively.

| SWE Scaffold | Base Model | Avg CPU Time (s) | Avg Inf. Time (s) | Avg $T_{IN}$ ($k$) | Avg $T_{OUT}$ ($k$) | Avg LLM Requests |
|---|---|---|---|---|---|---|
| AUTOCODEROVER | GPT-4o-mini | 14.7 | 84.3 | 276.7 | 14.2 | 39.5 |
| | Llama-3.3-70B | 9.5 | 91.7 | 416.1 | 14.5 | 38.3 |
| | Qwen3-32B | 34.5 | 105.6 | 55.5 | 20.4 | 14.7 |
| OPENHANDS | GPT-4o-mini | 94.7 | 41.4 | 737.4 | 1.7 | 34.1 |
| | Llama-3.3-70B | 142.8 | 64.1 | 897.1 | 4.9 | 30.8 |
| | Qwen3-32B | 102.5 | 99.1 | 335.4 | 16.7 | 19.5 |
| SWE-AGENT | GPT-4o-mini | 138.1 | 470.7 | 8143 | 24.4 | 181 |
| | Llama-3.3-70B | 15 | 25.9 | 193.7 | 3.2 | 29.1 |
| | Qwen3-32B | 22.4 | 225.2 | 440.2 | 41 | 35.5 |
| AGENTLESS | GPT-4o-mini | 889 | 141.1 | 29.4 | 27.7 | 82.3 |
| | Llama-3.3-70B | 454.8 | 166.2 | 63.9 | 32.4 | 80.0 |
| | Qwen3-32B | 727.9 | 209.4 | 26 | 41.4 | 83.1 |
| AGENTLESS-MINI | GPT-4o-mini | 0.7 | 4.6 | 35.2 | 1.7 | 2.0 |
| | Llama-3.3-70B | 0.8 | 11.3 | 37.5 | 4.0 | 2.0 |
| | Qwen3-32B | 0.8 | 49.4 | 34.1 | 9.3 | 2.0 |

trade-offs that are vital for anyone looking to build, deploy, or research performant and efficient AI software engineering systems. Lastly, we provide additional evaluation data in the Appendix.

## 5.1 OBSERVATION 1: SWE SCAFFOLD PERFORMANCE IS HIGHLY MODEL-DEPENDENT

Tables 1 and 2 highlight that the choice of base model has a stronger impact on outcome quality than SWE scaffold design alone. While SWE scaffolds such as OpenHands and SWE-Agent were previously among the top performers on SWE-bench, their performance drops significantly when paired with alternative LLMs. For example, SWE-Agent paired with Qwen3-32B achieves a resolution rate of 28% with 35.5 API calls and 440k input tokens, but this drops to just 10% when using GPT-4o-mini, despite requiring 181 calls and over 8.1 million input tokens—more than 18× the token cost. Similarly, OpenHands achieves 34% resolution and a respectable effectiveness under Token Budget (EuTB) of 22.7% with Qwen3-32B, but drops to 11.9% and its EuTB score collapses to a mere 6.8% when paired with GPT-4o-mini.

In contrast, AutoCodeRover, a simpler SWE scaffold, consistently delivers competitive performance. With Qwen3-32B, it achieves a resolution rate of 38% using only 14.7 API calls and 55.5K input tokens, which is much more efficient than its peers. Even with GPT-4o-mini, it maintains 12% resolution, outperforming OpenHands and SWE-Agent at the same model setting. These results

underscore the importance of LLM–scaffold synergy, where the effectiveness of a scaffold is tightly coupled with the reasoning capabilities and effectiveness under token budget (EuTB) of its underlying LLM.

Another key pattern is the performance of Agentless, which leads all SWE scaffolds in resolution rate (48% with Qwen3-32B), but at the cost of significant computation: it consumes an average of 83.1 API calls, 209.4 seconds of inference time, and 727.9 seconds of CPU time—among the highest in the table. This is due to its default design of generating numerous repair patches and tests, as well as executing regression and reproduction tests for each trial, trading off effectiveness for robustness.

These observations underscore the necessity of incorporating resource effectiveness metrics, as resolution rate alone may obscure the real cost of success.

## 5.2 Observation 2: High-Quality Reasoning Saves Tokens

Counter-intuitively, we observe that the reasoning model that consumes significantly more tokens per call can resolve issue instances with fewer total tokens. Table 1 shows that using a smaller, specialized "reasoning" model like Qwen3-32B was significantly more token-efficient than using a larger, general-purpose model like Llama-3.3-70B. Consider AutoCodeRover: with the specialized reasoning of Qwen3-32B, it achieves outstanding scores in every category: EuTB (37.1%), EuITB (33.1%), EuCTB (37.9%), and EuCB (37.0%). When using the larger, general-purpose Llama-3.3-70B model, all of its effectiveness scores are cut nearly in half.

The main reason is that, while a reasoning model may be more intensive per call, it enables the AI system to solve problems with fewer API calls (e.g., 15 calls for Qwen vs. 38 calls for Llama with AutoCodeRover). This reveals a critical insight: the primary driver of an AI system's total cost is the number of interactions, not the intensity of each one. A smarter model that can formulate a more concise strategy saves vast amounts of tokens and time by avoiding unnecessary back-and-forth communication.

## 5.3 Observation 3: Token Snowball: Cascading Amplification of Invalid Context

Figure 1 shows the relationship between the number of API calls per instance and the total number of input tokens consumed by that instance, comparing the three scaffolds: AutoCodeRover, OpenHands, and SWE-Agent. As the number of calls increases, input token consumption grows steadily in both cases—but crucially, this growth is roughly linear per call. This is because most agent frameworks today adopt a naïve memory accumulation strategy, where each new LLM response is appended to the next input prompt. While simple to implement, this leads to monotonic prompt growth, with each API call adding thousands of tokens even when the agent fails to make meaningful progress. We refer to this persistent design inefficiency as the *Token Snowball Effect*: even small per-call additions to the prompt accumulate into a large and often unnecessary context over long interaction trajectories. Snowball Effect creates not only excessive LLM cost and latency, but also a cognitive burden on the model: as the context window becomes cluttered with stale information, the LLM may lose focus on relevant cues, reducing reasoning quality and success rate, subsequently leading to more iterations and continuous Token Snowball Effect.

The "Token Snowball Effect" is particularly evident in SWE-Agent with GPT-4o-mini (Figure 1c), which frequently enters long, high-token trajectories without converging on a fix. By contrast, AutoCodeRover with Qwen3-32B (Figure 1a) tends to terminate earlier, often solving issues with fewer API calls and lower token usage overall. This suggests the necessity of developing better budget management strategies and advanced memory abstraction techniques. Better budget management strategies can balance scaffoldings' issue-resolving ability and the budget, stopping the task if the token snowball grows to an unacceptable level, but can barely resolve the issue, while better memory abstraction techniques can mitigate the growth of the token snowball.

## 5.4 Observation 4: Failing is Far More Expensive Than Succeeding

Table 3 compares the core performance metrics and resource usage between resolved (R) and unresolved (U) issue attempts. We can observe a trend that unresolved issues tend to consume more

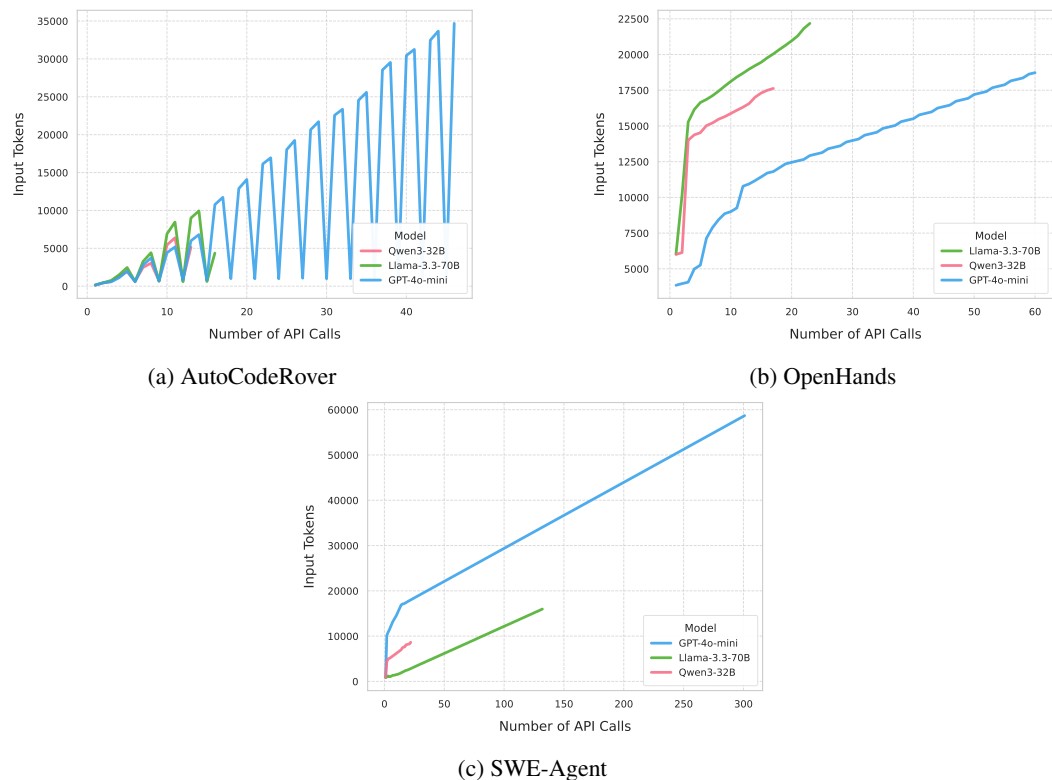

(a) AutoCodeRover

(b) OpenHands

(c) SWE-Agent

Figure 1: The relationship between the number of input tokens and the number of LLM calls. One SWE-Bench task instance is selected (*scikit-learn-14496*) from each of AutoCodeRover, OpenHands, and SWE-Agent as a showcase.

Table 3: Resource performance metrics comparison of AI systems across resolved (R) and unresolved (U) issues; mean across samples. *Inf.* - normalized inference time, $k$ - thousand.

| SWE Scaffold | Base Model | Total Time (s) | | CPU Time (s) | | Inf. Time (s) | | Total Tokens ($k$) | | LLM Requests | |
|---|---|---|---|---|---|---|---|---|---|---|---|
| | | U | R | U | R | U | R | U | R | U | R |
| AUTOCODEROVER | GPT-4o-mini | 109.5 | 21.4 | 16.3 | 2.9 | 93.0 | 25.4 | 328 | 25 | 43.5 | 10.0 |
| | Llama-3.3-70B | 122.7 | 45.9 | 10.8 | 6.1 | 111.9 | 39.8 | 540 | 147 | 45.0 | 20.9 |
| | Qwen3-32B | 173.5 | 85.4 | 52.2 | 5.5 | 121.4 | 51.2 | 91 | 51 | 16.1 | 12.4 |
| OPENHANDS | GPT-4o-mini | 142.9 | 86.0 | 99.9 | 55.6 | 42.9 | 30.3 | 775 | 472 | 34.2 | 35.2 |
| | Llama-3.3-70B | 238.9 | 79.0 | 164.5 | 56.0 | 74.4 | 23.0 | 1068 | 238 | 34.9 | 14.3 |
| | Qwen3-32B | 237.4 | 131.8 | 118.0 | 72.2 | 119.4 | 59.6 | 424 | 211 | 22.5 | 13.6 |
| SWE-AGENT | GPT-4o-mini | 658.0 | 167.2 | 146.8 | 59.9 | 51.1 | 107.3 | 8867 | 1865 | 192.0 | 81.6 |
| | Llama-3.3-70B | 43.9 | 19.0 | 15.8 | 8.4 | 28.0 | 10.6 | 220 | 28 | 31.4 | 12.7 |
| | Qwen3-32B | 274.8 | 177.8 | 22.8 | 21.4 | 251.9 | 156.4 | 519 | 383 | 37.4 | 30.4 |
| AGENTLESS | GPT-4o-mini | 998.4 | 1120.7 | 852.5 | 993.0 | 145.9 | 127.7 | 58 | 54 | 81.8 | 83.9 |
| | Llama-3.3-70B | 653.0 | 538.5 | 479.8 | 390.6 | 173.3 | 147.9 | 101 | 83.7 | 78.5 | 83.8 |
| | Qwen3-32B | 1037.3 | 829.0 | 774.3 | 677.7 | 263.0 | 151.3 | 79.3 | 54.4 | 82.6 | 83.5 |
| AGENTLESS-MINI | GPT-4o-mini | 5.2 | 5.4 | 0.64 | 0.78 | 4.5 | 4.6 | 33.5 | 44 | 2.0 | 2.0 |
| | Llama-3.3-70B | 12.4 | 4.3 | 0.76 | 0.54 | 11.6 | 3.8 | 39.4 | 31.6 | 2.0 | 2.0 |
| | Qwen3-32B | 62.6 | 18.3 | 0.76 | 0.77 | 61.8 | 17.5 | 46 | 36.8 | 2.0 | 2.0 |

resources on average, including longer inference time, higher token usage (Figure 2), and more execution steps, indicating inefficiencies in failure cases. Take SWE-Agent with GPT-4o-mini as an example: a failed attempt consumes over 8.8 million tokens and 658 seconds, while a successful one requires just 1.8 million tokens and 167.2 seconds. That's more than a 4x increase in both inference time and token cost when the agent is off track. The same pattern holds elsewhere—OpenHands with Llama-3.3-70B takes 238.9 seconds on average when it fails, compared to 79 seconds when it

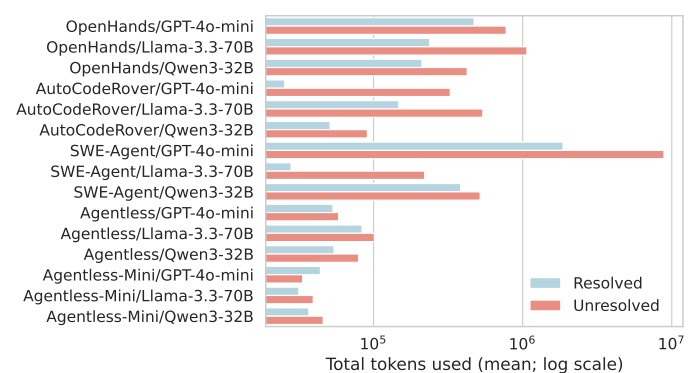

Figure 2: Total input and output tokens used for resolved and unresolved attempts; mean across instances. On average, unresolved attempts utilized more tokens than resolved attempts.

succeeds. Even efficient scaffolds like AutoCodeRover aren't immune: a failure often takes nearly three times as long as a success.

This "fail expensively" behavior highlights a critical capability gap: the lack of futility detection. When AI systems are on a productive path, they often reach a solution efficiently. But when they're stuck, they enter expensive, repetitive loops, consuming massive amounts of compute until an external budget limit is reached. These hidden costs pose a serious challenge for real-world deployment. Future research may explore how scaffolds can become progress-aware, identifying signals of stagnation—analogous to code smells in software engineering—and learning to abort or redirect unproductive trajectories before problem-solving spirals out of control.

## 6 DISCUSSION AND CONCLUSION

In this work, we systematically explored an evaluation paradigm for AI software engineering systems that extends beyond the primary metric of resolve rate. Our analysis of five popular scaffolds in combination with three popular LLMs surfaced several key observations with direct implications for future research. We observed that an AI system's success is not determined by the scaffold alone but by its synergy with the base model, which is crucial for achieving high performance efficiently. We also identified a clear trade-off between effectiveness under cost budget (tokens) and effectiveness under time budget (latency), directly impacting both a project's budget and the feasibility of large-scale Reinforcement Learning where low latency is critical for training. Lastly, we observed systemic issues like the "Token Snowball" effect and, most critically, a tendency for AI systems to "fail expensively" where AI system get stuck in resource-intensive loops, hindering their real-world use but also posing a significant barrier to effective RL training.

It is important to note that this work is intended to provide an introduction offering insights into the multifaceted evaluation of AI system effectiveness. It is not exhaustive with respect to all scaffold-model combinations. To show an example for software practitioners in gaining a deeper understanding when selecting approaches with affordable resources and time consumption, we adopted stratified sampling to select a representative subset from SWE-bench-Verified and derived conclusions with clear statistical interpretation. We aim to evaluate more AI systems in the future with the help of the broader research community by releasing our code, data, and hosting a public leaderboard.

It is crucial to highlight that while our experiments were conducted on the SWE-bench benchmark, the insights gained are not confined to software engineering. The systemic issues we identified—such as the "Token Snowball" effect and the pattern of "expensive failures"—are likely fundamental challenges for any autonomous AI system designed to solve complex, multi-step problems in any domain. In light of our findings, we believe that a lightweight framework that avoids expensive failure modes is not only a more practical tool today but also a more "evolvable" foundation for the reinforcement learning approaches that will shape the future of AI in software engineering.

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

# 7 APPENDIX

## 7.1 COSTS USED FOR EFFECTIVENESS UNDER COST BUDGET (EuCB) CALCULATIONS

To calculate the EuCB, we utilized USD-per-million-tokens costs found on *openrouter.ai* as of July 11, 2025 and picked the cheapest provider: $0.15 input and $0.6 output for GPT-4o-mini, $0.38 input and $0.12 output for LLama-3.3-70B-Instruct, $0.10 input and $0.30 output for Qwen3-32B.

## 7.2 ADDITIONAL RESOURCE EFFICIENCY EVALUATION

594
595
596
597
598
599
600
601
602
603
604
605
606
607
608
609
610
611
612
613
614
615
616
617
618
619
620
621
622
623
624
625
626
627
628
629
630
631
632
633
634
635
636
637
638
639
640
641
642
643
644
645
646
647

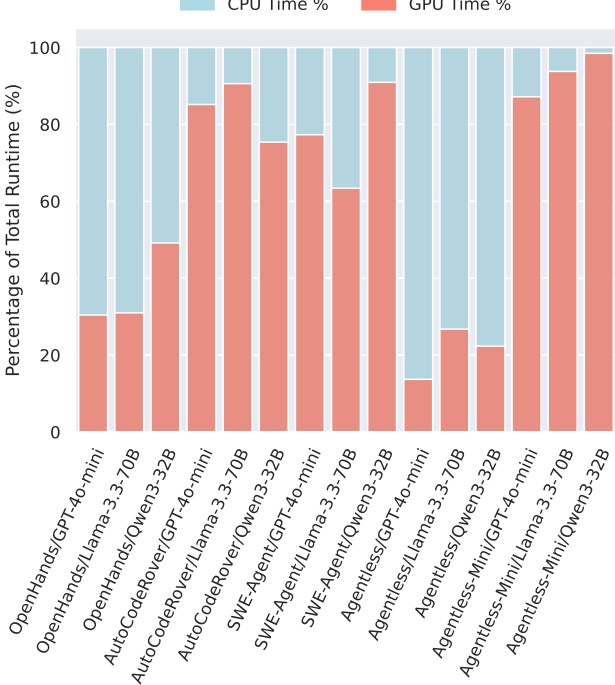

Figure 3: Proportion of runtime spent on CPU or inference (GPU) operations. Scaffolds AutoCodeRoder, SWE-Agent, and Agentless-Mini generally spent the most of the total runtime performing inference operations, meanwhile OpenHands and Agentless spent majority of runtime on CPU tasks (e.g., running tests). High inference time for Agentless-Mini is due to scaffold's two-stage design with nearly no intensive CPU-bound computation. In case of Agentless, CPU time dominates as it involves execution of numerous reproduction and regression tests.

12

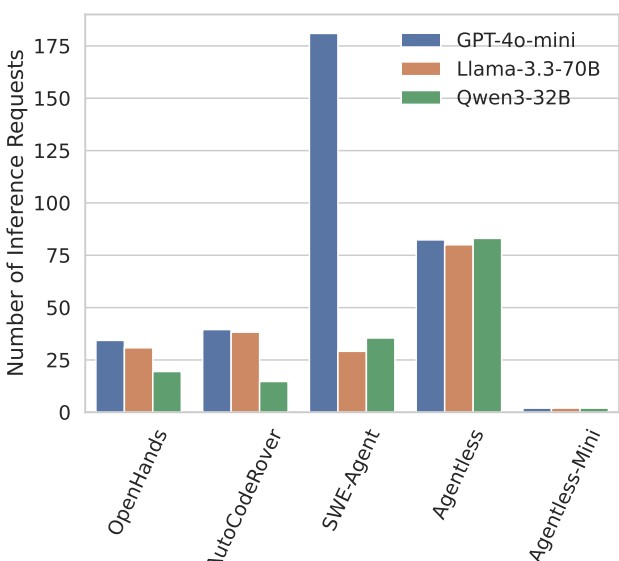

Figure 4: Number of LLM requests per scaffold and model combinations; mean across all instances in select set. The spike in number of inference requests for GPT-4o-mini with SWE-Agent scaffold was due to significantly lower USD-per-tokens costs compared to other models, allowing for more attempts before the cost budget was reached. With the exception of SWE-Agent, GPT-4o-mini and Llama-3.3-70B had similar number of API inference requests across the scaffolds. At the same time, Qwen3 generally had similar to lower number of requests, particularly for OpenHands and AutoCodeRover scaffolds, potentially due to more effective responses provided by Qwen3.

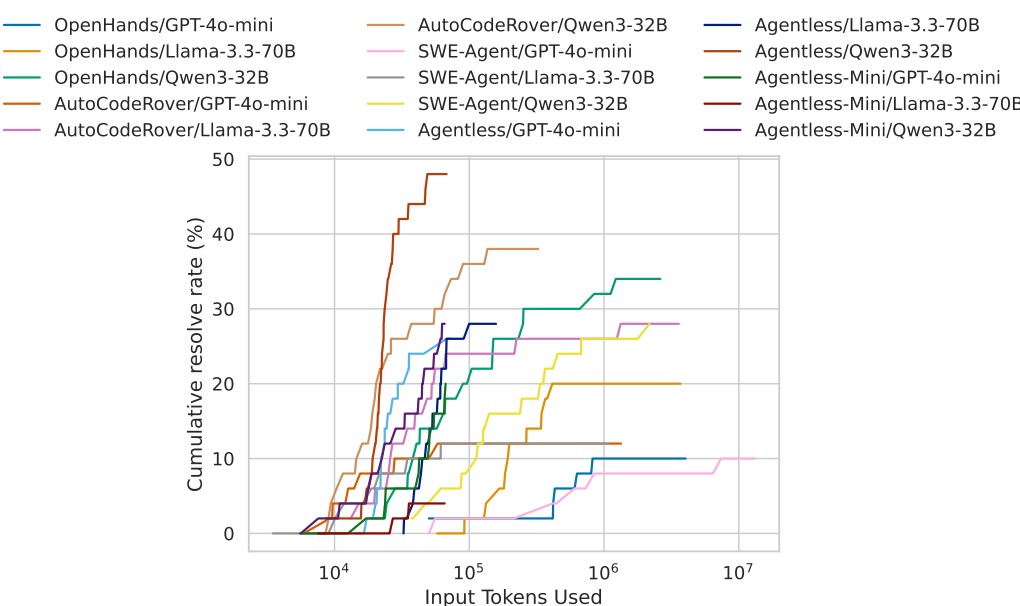

Figure 5: Resolve rate given a per-issue input token usage. This and Figure 6 curves were used for calculation of EuTB metric. Interactive version of this plot will be provided on the leaderboard website.

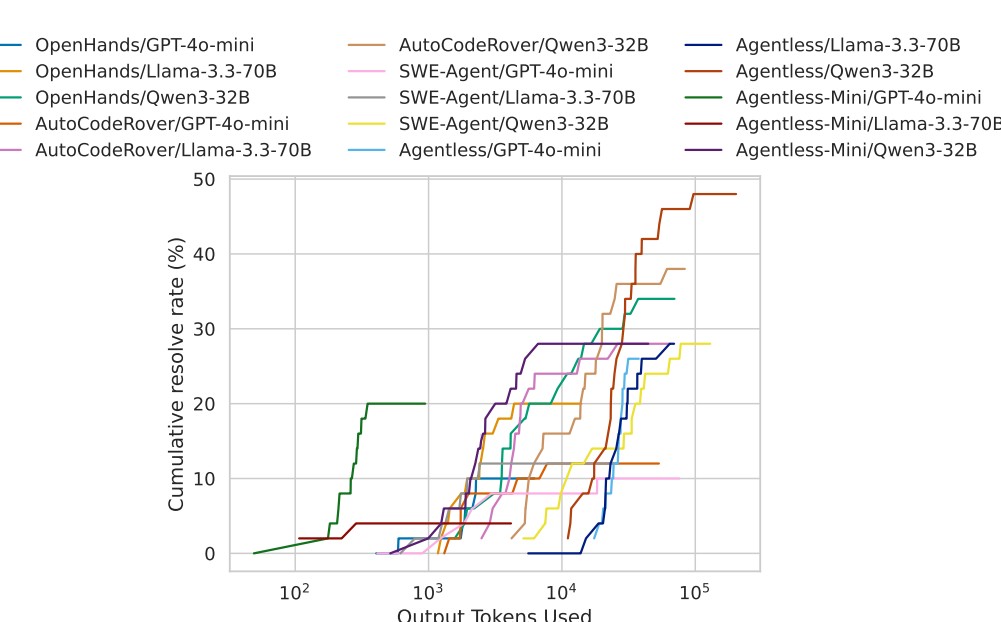

Figure 6: Resolve rate given a per-issue output token usage. This and Figure 5 curves were used for calculation of EuTB metric. Interactive version of this plot will be provided on the leaderboard website.

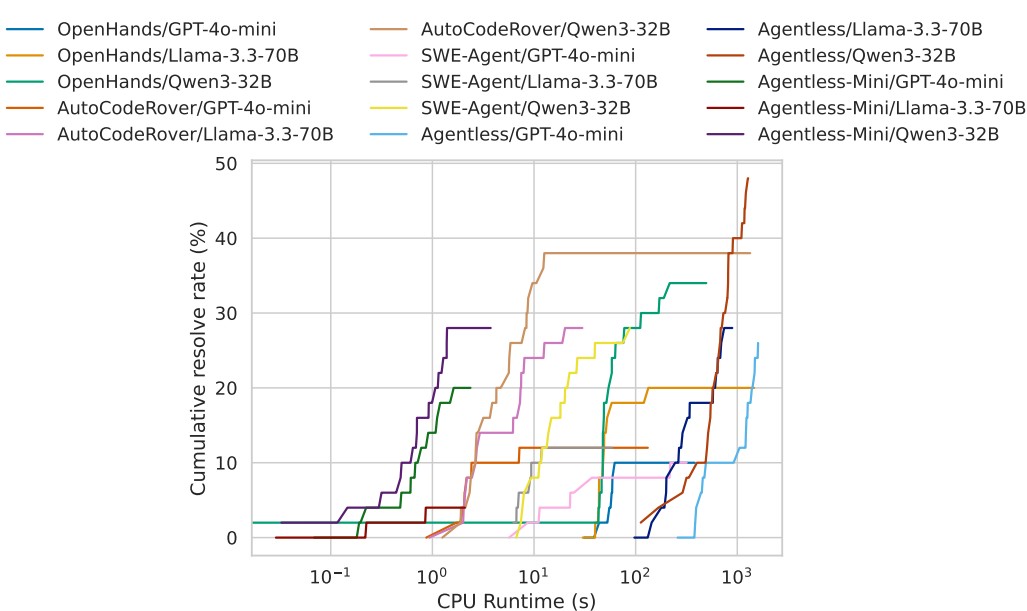

Figure 7: Resolve rate given a per-issue CPU runtime, showing how long a task spent on CPU-bound processing. Interactive version of this plot will be provided on the leaderboard website.