# OpenReview forum: "SWE-Effi: Re-Evaluating Software AI Agent System Effectiveness Under Resource Constraints"
_ICLR.cc/2026/Conference — Submitted to ICLR 2026_

### Official Review · Reviewer_HZM9 · 2025-10-29

**Soundness:** 2
**Presentation:** 3
**Contribution:** 1
**Rating:** 2
**Confidence:** 4

**Summary:**

The paper attempts to evaluate SWE Agent systems beyond measuring their resolve rate. They quantify other properties of their behavior which may be relevant to how these systems need to be used in the real world. They specifically focus on cost and run-time, comparing how different agent scaffolds combined with different models perform . The paper highlights the importance of model and scaffold interaction as well as drawbacks and failure modes of current systems such as "failing expensively".

**Strengths:**

1. The paper takes a novel approach to evaluating SWE Agents, assessing their feasibility by evaluating them not only on resolve rate but also resources used (in terms of computation time and money).

2. The paper highlights how current systems "fail expensively", opening up directions to make them more usable by having early failure detection.

**Weaknesses:**

1. Limited scope of evaluation: While the metrics proposed in the paper are insightful, they provide limited insights into how useful the agent might be in real world applications.

2. Limited analysis of differences in agent behaviour. The does not go beyond showing that different scaffold result in different performance. It is unclear for example why AutoCodeRover does better than SWE-Agent, or why Qwen3-32B does better than GPT-4o-mini. Without understanding this, the results of the paper's evaluation will be of limited help to model and agent designers.

3. The results presented in the paper are using weaker than SOTA models along with a very low cost limit of $1. The choice of $1 seems arbitrary. Furthermore, the performance of SOTA models with (presumably) higher cost limits (as observed on https://www.swebench.com/) are significantly higher (upto 70%) compared to the evaluations in the paper. While the results in low cost regime are interesting, the behaviour of SOTA models may be significantly different and hard to infer from the current set of experiments.

4. The paper discusses the problem of token snowballing. While it is a reasonable hypothesis that history/memory management strategies for agents may result in better performance, such a setting is not explicitly evaluated, since there maybe a tradeoff between forgetting in the memory and maintaining full history. Furthermore, model providers already implement server-side caching which may mitgate this issue:  https://platform.openai.com/docs/guides/prompt-caching

Overall, my major reason for recommending rejection is the limited scope of evaluation, analysis and concerns related to experimentation. While the ideas presented in the paper are well motivated, the contributions do not seem for acceptance.

**Questions:**

1. I see that. GPT-4o with SWE-Agent makes ~4x as many calls as other combinations. Do the authors know why this is the case? Is there something particularly incompatible about this combination?

---

### Official Review · Reviewer_D9D4 · 2025-10-31

**Soundness:** 2
**Presentation:** 3
**Contribution:** 2
**Rating:** 2
**Confidence:** 4

**Summary:**

SWE-Effi is a set of new metrics to evaluate performance of coding agents (and potentially other AI agents as well) against resource usage. The paper evaluates some SLMs on different agent scaffolds on a subset of swebench-verified benchmark. By measuring the effectiveness within resource bounds, they draw insights about performance tradeoffs for different configurations.

**Strengths:**

- The authors make a valid point that coding agents should be measured on their effectiveness under resource constraints. They define different resource constraints targeting CPU and inference time, API calls, etc.
- The paper evaluates five agent scaffolds and three models on a subset of swebench-verified. They observe that the choice of scaffold and the synergy between the model and the scaffold are among the factors that affect the agent effectiveness.

**Weaknesses:**

- The central premise of the paper is valid and useful, but in the current form, the paper's contributions are not strong enough for ICLR.
- The experiments are conducted only with SLMs and no LLMs are tested, whereas in practice, developers heavily use LLMs. The evaluation is conducted on just one benchmark, swebench-verified, and further only on a subset of 50 tasks. Both of these are important limitations.
- I also found the analysis part of the paper lacking. The paper should provide deeper insights into what choices in scaffolds play a role in the seen performance (e.g., what causes or prevents the token snowball effect).
- Typo: Table 1, EuCB column, the max is 47.1 but 37.0 is highlighted as the max.

**Questions:**

- How do the paper's observations generalize to different datasets and LLMs?
- What are the key takeaways for LLM and agent designers to optimize their performance within resource constraints?

---

### Official Review · Reviewer_D1S9 · 2025-10-31

**Soundness:** 2
**Presentation:** 3
**Contribution:** 2
**Rating:** 4
**Confidence:** 4

**Summary:**

This paper introduces SWE-Effi, a multi-dimensional evaluation framework for assessing AI software engineering systems beyond traditional accuracy metrics. The authors evaluate 5 popular scaffolds (AutoCodeRover, SWE-Agent, OpenHands, Agentless, Agentless-Mini) paired with 3 LLMs (GPT-4o-mini, Llama-3.3-70B, Qwen3-32B) on 50 issues from SWE-bench-Verified. The framework introduces four effectiveness metrics based on Area Under the Curve: token budget (EuTB), cost budget (EuCB), CPU time budget (EuCTB), and inference time budget (EuITB).

Key findings include: (1) scaffold performance is highly model-dependent, (2) better reasoning models can be more token-efficient overall, (3) "token snowball" effect from context accumulation, and (4) failures consume significantly more resources than successes. While there are good ideas in the paper, there are some concerns such as issues in methodology, clarity of presentation, etc. that I call out below in Weaknesses/questions to authors.

**Strengths:**

1. Resource efficiency in AI systems is understudied; addressing the "resolve rate tunnel vision" is valuable.

2. Multi-dimensional metrics (tokens, cost, CPU time, inference time) provide richer understanding than single metrics

3. Model-scaffold synergy and expensive failures have practical implications

4. Findings on effectiveness vs. latency tradeoffs are relevant for RL training

**Weaknesses:**

The paper suffers from major methodological flaws: an inadequate sample size (N=50) with ±13% confidence intervals undermines quantitative claims and leaderboard validity; the normalized inference-time model (Equation 1) is trained on one model yet applied across architectures without validation, leaving 21% unexplained variance.

The “expensive failures” analysis confounds difficulty with cost and ignores contradictory evidence (e.g., Agentless). The “token snowball effect” is overstated with only anecdotal examples and no systematic analysis. Experimental modifications (parallelism disabled, SWE-Agent $1 cap, API changes) reduce fairness and practical relevance. These issues together tend to compromise the reliability and generalizability of the findings.

**Questions:**

1. N=50 gives ±13% CI, how is this significant? Please provide more analysis.

2. Equation 1 is trained on GPT-4o-mini but applied across architectures/providers. Why assume same linearity? Report $r^2$ for others or use actual wall-clock times. How does this uncertainty affect EuITB rankings?

3. Agentless shows resolved issues cost **more** CPU time, contradicting your claim. Why? Also, why no control for issue difficulty?

4. Why disable parallelism and cap SWE-Agent at $1 while others use defaults? How do these changes affect benchmark relevance?

5. It would be good to articulate the novelty angle beyond prior work (e.g., MemPrompt)?

6. Also, why only one example per scaffold? Can you provide systematic evidence across tasks?

---

### Official Review · Reviewer_Z1VA · 2025-11-01

**Soundness:** 3
**Presentation:** 2
**Contribution:** 2
**Rating:** 4
**Confidence:** 4

**Summary:**

This paper proposes several evaluation scenarios and metrics for software engineering agents in a resource-constrained setting, including token-constrained, budget-constrained, cpu-time constrained, and inference-time constrained settings. This provides a more comprehensive picture for AI SWE practitioners to choose solutions for different use cases.

**Strengths:**

1. Rigorous measurement setup: using normalized inference time that is computed with regression model trained on real API calls, avoiding instability in actual API call inference time and making comparison fairer.

2. Interesting and sometimes surprising findings, such as reasoning models that use more tokens per step can actually save tokens by being smart.

3. Very comprehensive evaluation across 5 agents and 3 models, providing useful information for the community.

**Weaknesses:**

1. Lack of technical novelty. All metrics seem to be intuitive, which is good. But similar yet smaller analyses have been done in other coding agent papers.

**Questions:**

1. How do the effectivness metrics differ across different tasks? Are some tasks costing more tokens than others? What are some signs indicating those hard-to-solve issues?

2. Tiny issue: Reference format. Use \citep for certain references (for example those on Line 222).

---

### Meta-Review · Area_Chair_vV6f · 2026-01-06

**Summary:**

The paper presents SWE-Effi, a framework for evaluating software-engineering agents under resource constraints, using AUC-style metrics for token usage, cost, CPU time, and latency. It tests 5 agent scaffolds with 3 models on 50 SWE-bench-Verified issues and reports findings such as scaffold–model interaction, a “token snowball” effect, and resource-intensive failures.

Although evaluating agents under resource limits is a meaningful goal, reviewers raise major concerns that weaken confidence in the results and the paper’s suitability for acceptance. Key issues include: (i) methodological reliability, notably the small sample size (N=50), large uncertainty, and an inference-time normalization model trained on one model but applied broadly; (ii) limited scope, relying on one benchmark and a 50-issue subset and emphasizing weaker-than-SOTA settings; (iii) fairness and practicality, including disabled parallelism and a $1 cost cap; and (iv) limited depth of analysis, with claims such as “token snowball” and “expensive failures” viewed as overstated or weakly supported.

**Reviewer Concerns:**

* **Methodological validity & statistical strength:** N=50 yields wide confidence intervals; the normalized inference-time model lacks cross-model validation and leaves substantial unexplained variance.
* **Scope & representativeness:** Only SWE-bench-Verified and a 50-task subset are used; results rely on weaker models and a low cost cap, limiting generalization.
* **Fairness & practical relevance:** Modified experimental settings (e.g., disabled parallelism, $1 cap) reduce realism and fairness.
* **Insufficient analysis depth:** Limited explanations for why scaffolds or models differ; effects like “token snowball” are insufficiently substantiated.
* **Limited novelty:** Metrics and observations are viewed as incremental relative to prior work.
* **Generalization gaps:** No LLM evaluation; unclear applicability to stronger models or other datasets.

Because no rebuttal addresses these issues through added experiments or revised methodology, these concerns remain unresolved.

**Reviewer Scores:**

The author did not provide a rebuttal, so the reviewer scores are likely to remain unchanged.

---

### Decision · Program_Chairs · 2026-01-26

Reject